# Observation of Josephson-like Tunneling Junction Characteristics and Positive Magnetoresistance in Oxygen Deficient Nickelate Films of Nd_0.8_Sr_0.2_NiO_3−*δ*_

**DOI:** 10.3390/ma14247689

**Published:** 2021-12-13

**Authors:** Gad Koren, Anna Eyal, Leonid Iomin, Yuval Nitzav

**Affiliations:** Department of Physics, Technion—Israel Institute of Technology, Haifa 32000, Israel; anna.eyal@gmail.com (A.E.); leo@physics.technion.ac.il (L.I.); yuval.nitzav@gmail.com (Y.N.)

**Keywords:** tunneling, superconductivity, thin films, magnetoresistance, 74.50.Br, 74.78.Aw, 73.50.Ah, 73.43.Qt

## Abstract

Nickelate films have recently attracted broad attention due to the observation of superconductivity in the infinite layer phase of Nd${}_{0.8}$Sr${}_{0.2}$NiO${}_{2}$ (obtained by reducing Sr doped NdNiO${}_{3}$ films) and their similarity to the cuprates high temperature superconductors. Here, we report on the observation of a new type of transport in oxygen poor Nd${}_{0.8}$Sr${}_{0.2}$NiO${}_{3-\delta }$ films. At high temperatures, variable range hopping is observed while at low temperatures a novel tunneling behavior is found where a Josephson-like tunneling junction characteristic with serial resistance is revealed. We attribute this phenomenon to coupling between superconductive (S) surfaces of the grains in our Oxygen poor films via the insulating (I) grain boundaries, which yields SIS junctions in series with the normal (N) resistance of the grains themselves. The similarity of the observed conductance spectra to the tunneling junction characteristic with Josephson-like current is striking, and seems to support the existence of superconductivity in our samples.

## 1. Introduction

The recent discovery of superconductivity in Nd${}_{0.8}$Sr${}_{0.2}$NiO${}_{2}$ thin films [1] has brought up research of the Nickelates to the front of condense matter physics [2]. These materials are interesting since they are similar in structure and electronic properties to the cuprate family which is well known for its high temperature superconductivity. It is expected that understanding the similarities and differences between these two families of materials will complement each other and illuminate the mechanism of superconductivity in the cuprates which is still under debate. The phase diagram of the Nd${}_{0.8}$Sr${}_{0.2}$NiO${}_{2}$ films and more of their basic properties are given in [3]. Superconductivity was observed also in Pr${}_{0.8}$Sr${}_{0.2}$NiO${}_{2}$ thin films [4] but in both this study and Ref. [1] the films thickness ranged between 5 and 12 nm only. Thicker films of this kind showed absence of superconductivity in bulk form [5,6] and raised the question whether the observed superconductivity in the few nm thick films is an interface effect originating in strains with the substrate. The way in which the Nd${}_{0.8}$Sr${}_{0.2}$NiO${}_{2}$ thin films were prepared involved laser deposition of the perovskite Nd${}_{0.8}$Sr${}_{0.2}$NiO${}_{3}$ films first, which were then reduced by a chemical annealing process with CaH${}_{2}$ that changed them to the superconductive infinite layer phase [1]. Similar films were fabricated successfully by other groups using the same process [7,8,9], but it was clear from the beginning that this reduction process is not just simple extraction of oxygen from the films since it involved also a major change of their structural phase. In view of this, a few groups tried to reproduced Ref. [1] results by physical extraction of oxygen from the precursor Nd${}_{0.8}$Sr${}_{0.2}$NiO${}_{3}$ films by annealing at elevated temperatures under vacuum or with low oxygen pressure. Unfortunately, this led to even more insulating films and no change of phase to the superconductive infinite layer one was obtained [5,6]. In the present study, we also used physical reduction of the precursor insulating films of Nd${}_{0.8}$Sr${}_{0.2}$NiO${}_{3}$ and investigated their properties. In particular, annealing under 10 mTorr oxygen increased the films resistivity at low temperatures by two order of magnitudes but did not change their structural phase. These films showed variable range hopping (VRH) transport in 2D versus temperature with deviation due to tunneling conductance below about 4 K. In view of our results, we make the conjecture that this deviation toward increasing resistance originates in gap opening and tunneling between superconductive (S) surfaces of the grains in the films via the insulating (I) grain boundaries. Thus accordingly, our films can be visualized as comprised of a network of SIS tunneling junctions connected in series with additional serial resistance of the normal (N) grains themselves.

## 2. Experimental

### 2.1. The Surface Morphology of the Films

Standard pulsed laser ablation deposition of the films from stoichiometric Nd${}_{0.8}$Sr${}_{0.2}$NiO${}_{3}$ ceramic target was used (see methods). The resulting films were shiny black, well crystallized and very smooth as can be seen by the AFM images of Figure 1 for the two films deposited at 600 and 650 ${}^{\circ }$C.

### 2.2. X-ray Characterization of the Films

Figure 2 exhibits X-ray diffraction results of our as deposited films on (100) SrTiO${}_{3}$ wafers under various substrate temperatures. It shows that the Nd${}_{0.8}$Sr${}_{0.2}$NiO${}_{3}$ perovskite phase was obtained, except for the too low 450 ${}^{\circ }$C deposition which can be considered as background for the other data. We focus on two regions near the most prominent (002) and (004) peaks of the films as depicted in (b) and (c), respectively. The peak at (001) was weak and the one at (003) was absent. The measured 2θ width at half maximum (FWHM) of the un-split (004) peaks of the different films is about 0.78${}^{\circ }$ which is approximately equal to twice the split of the (400) STO peaks (0.36${}^{\circ }$) resulting from the $K\alpha_{1}$ and $K\alpha_{2}$ Copper lines as seen in (c). For the (002) peak of the films as seen in (b), the measured width is 0.33${}^{\circ }$ which is about three times the split of the (200) peak of STO (0.12${}^{\circ }$). Comparing our data to that of Li et al. [1] reveals two important differences: One is the much larger width of the (002) peak in Ref. [1] of about 1${}^{\circ }$ FWHM as compared to ours of 0.33${}^{\circ }$ FWHM even though ours includes the $K\alpha_{2}$ Copper line contribution. This indicates a much better crystallization of the films in the present study. It might still have been advantageous for Li et al. to have this poorer crystallization since the objective was to obtain the infinite layer phase of Nd${}_{0.8}$Sr${}_{0.2}$NiO${}_{2}$ for which the chemical reduction of poorer precursor was easier. A second difference is that the (001) and (002) peaks in [1] were of comparable intensity while, in our data, the (001) peak is much weaker than the (002) one. Li et al. also say that their (003) film peak is not visible owing to its low intensity which is also what we observed. Generally, however, the intensity of the even peaks in a perovskite [(002) and (004)] is much higher than that of the odd peaks [(001) and (003)], such as, for instance, in STO. Thus the robust (001) peak in Ref. [1] is puzzling, but it might be associated with the much thinner strained films involved.

Another relevant issue is the small minority phases observed in Figure 2d. These were identified as due to various Nickel oxides Ni${}_{x}$O${}_{y}$, of which we marked in (d) NiO, Ni${}_{2}$O${}_{3}$, and Ni, but other mixed stoichiometries are also possible as the broad, unresolved multiple peaks near 44${}^{\circ }$ indicates. All these peaks are shifted to lower angles by about 0.7${}^{\circ }$ compared to their location in powder diffractometry due to compressive strains with the STO substrate and the Nd${}_{0.8}$Sr${}_{0.2}$NiO${}_{3}$ films. For instance, the lattice constant of cubic NiO (0.4177 nm) is larger than that of the STO wafer (0.3904 nm) and the present NSNO8 film (0.3838 nm), thus compressive epitaxial strain in the a–b plane of the NiO elongates its c-axis (to 0.4236 nm here) and lowers the diffraction angle to where it appears in (d). To estimate the percentage of these minority phases in our films as seen in (d), we compare their peak heights (between 15 and 75 counts) to that of the (002) peak of NSNO8 in (b) (5900 counts), and obtain about 0.3–1.3%. Generally, one would ignore such small quantities of secondary minority phases, but since it is most likely that these phases reside on the surface of the films and in their grain boundaries (due to zone refining), they are very relevant for the present study where the transport in our films is strongly affected by these regions. In particular, as we shall see in the following, the magnetic properties of these phases [10,11,12,13] could play a role in explaining our transport results.

### 2.3. Physical Reduction of the Films

As mentioned before, physical reduction of our films was carried out rather than the chemical reduction with CaH${}_{2}$ used previously [1,3,4,7,8,9]. Three annealing procedures were used all at the same 200 ${}^{\circ }$C temperature. The first two under vacuum for 24 h and for 1 h, and the third one under 10 mTorr of Oxygen flow for 1 h and cooling under the same pressure. The first two annealing processes yielded very resistive films with several tens of MΩ resistance. The third annealing process was chosen since deposition (at 600 ${}^{\circ }$C) by Zhou et al. [6] at the same Oxygen pressure did not yield the Nd${}_{0.8}$Sr${}_{0.2}$NiO${}_{3}$ phase at all. For our NSNO8 film deposited also at 600 ${}^{\circ }$C, the low annealing temperature preserved the Nd${}_{0.8}$Sr${}_{0.2}$NiO${}_{3-\delta }$ phase, but increased the film resistance at low temperatures by two orders of magnitude. We note also that all annealing processes were reversible, and once reannealed at 200 ${}^{\circ }$C and 100 mTorr Oxygen pressure the films returned almost to their original as deposited state. Plots of some resistance versus temperature results of the various films with and without annealing is given in Figure 3. We note here that thinner films of 20 and 35 nm thickness as compared to the 70 nm thick films used presently, were too resistive to allow for a manageable dynamic resistance range of the transport measurements after reduction as shown in Figure 3. It should be added also that the Nickelate structure is very similar to that of the cuprates in the sense that both have Ni oxide and Cu oxide planes with their oxygen easily removed or replenish with Oxygen poor or rich annealing processes, respectively. One of the earliest studies of our group demonstrating these effects in the cuprates is given in Ref. [14], and we, therefore, assumed the same behavior also in the Nickelates.

### 2.4. The Measuring Systems and Contacts

Two measuring systems were used in the present study: One a physical properties measurement system with a closed cycle refrigerator and a magnet of up to 14 T (PPMS and DynaCool of Quantum Design) and the other a home made transport measurements probe inside a cryogenic dewar with liquid He and a magnet of up to 8 T (Teslatron of Oxford Inst.). The probe of the latter system had an array of 4 × 10 gold-coated spring loaded tips pressed onto the films for a 10 four-probe measurements on different location ($R_{i}$ for i=1 to 10) on the 10×10 mm${}^{2}$ wafer. Due to a problem with this probe, the high temperature readings were unreliable and, therefore, the corresponding data in Figure 3 was truncated above 150 K. Nevertheless, the Teslatron system was much more sensitive, versatile and controllable than the PPMS system, in particular in IVC and conductance spectra measurements at low temperatures and therefore all these measurements were carried out on it. To minimize the contact resistance four parallel silver paste bands for the four probe measurements were applied to the films as shown in the inset to Figure 3. In the PPMS four contacts were first wire-bonded to the film and then the Ag paste bands were applied to overlap each bonding spot and extend the contacts along the wafer. In the Teslatron probe, 10 measurements were performed simultaneously on 10 different areas of the film in such a way that the spring loaded contact tips were touching each band for every set of four contacts (three such sets are marked by red circles in the inset to Figure 3).

## 3. Results and Discussion

### 3.1. Variable Range Hopping in 2D

As can be seen from Figure 2 and Figure 3, all three films are c-axis oriented, have the perovskite Nd${}_{0.8}$Sr${}_{0.2}$NiO${}_{3}$ phase and show insulating behavior at low temperatures, more so when reduced to extract Oxygen from them. As no metallic behavior is observed in the R vs. T results of Figure 3, one has to conclude that the dominant part of the resistance is determined by the insulating grain boundaries. Since the NSNO8 film deposited at 600 ${}^{\circ }$C still had the perovskite phase as reported also by other groups [1,7,8], we chose to focus on it. Moreover, its annealing under 10 mTorr O${}_{2}$ flow at 200 ${}^{\circ }$C for 1 h yielded a convenient resistance range at low temperatures for IVC and conductance measurements. To find out what is the conductance mechanism of the insulating as deposited film and the reduced film, we plot in Figure 4a,b their resistance on a log-scale versus $T^{-1/3}$. The expected behavior on such a plot should be linear if variable range hopping (VRH) in 2D occurs [15,16]. As one can see in Figure 4a, this is actually so over one order of magnitude of resistance change for the reduced film, but the linearity range hardly covers a factor of two change in resistance for the as deposited film. In addition, the resistance data of the latter as measured in the PPMS were quite noisy and allowed for a straight line to be drawn through the data, as seen in (a). Measurement on three different areas (R2, R4, and R8) of the same film on the Teslatron system was much less noisy (see Figure 4b), and one can see that the linear range in R in this case is even smaller (about 100 Ω only), if at all. For comparison, we also tried to fit the NSNO8an1 resistance data to two additional VRH models, one of Coulomb coupling between conducting grains (log R vs. $T^{-1/2}$) as seen in (c), and the other of VRH in 3D (log R vs. $T^{-1/4}$) as depicted in (d). Linearity is observed in all models, but its temperature range is largest in (a) (37 to 4.6 K). The corresponding linearity range in (c) is 2 to 11 K, and in (d) 4 to 27.2 K. If the resistance range where linearity is observed is the important factor in determining which of the VRH models applies, then (c) is the most likely model. However, we believe that the upturn from linear behavior below about 4 K in the annealed film in Figure 4a is a real effect which indicates decreased conductance with decreasing temperature. As we shall see in the following, this is due to tunneling conductance and gap opening which lower the number of states to which the electrons can tunnel and therefore increase the film resistance compared to the VRH behavior.

### 3.2. Tunneling Conductance in the NSNO8 Films

Figure 5 depicts typical conductance spectra at various temperatures on two different areas of the as deposited NSNO8 and the oxygen deficient NSNO8an1 films. As we have seen before in Figure 4a, transport in the annealed NSNO8an1 film occurs via VRH in 2D in an intermediate temperature range, while at low temperatures we proposed that tunneling takes over. Figure 5 actually shows this tunneling conductance behavior with a narrow gap and a wide gap. The narrow gap is well developed at 1.8 K and up to at least 5 K in both films. This gap fills-in with increasing temperature and disappears above about 20 K, which is very similar to superconductive gap behavior. However, Figure 5a shows the conductance of the as deposited film is insensitive to a magnetic field of 1 T at 1.8 K. The same is true for the annealed film under 4 T field (will be shown later). These findings are at odds with the existence of conventional spin-singlet superconductivity in our films, where suppression of the zero bias conductance under field is expected. The observed insensitivity to magnetic field therefore might indicate that unconventional superconductivity, such as same-spin triplet superconductivity is at the origin of the observed gap in Figure 5 [17]. The presence of a small amount of ferromagnetic NiO in our films as seen in Figure 2d and also found by He et al. [18] in similar polycrystalline Sm${}_{0.8}$Sr${}_{0.2}$NiO${}_{3}$, could facilitate triplet superconductivity as found in other ferromagnetic-superconductor systems [19,20]. Actually, in Ref. [17] Matthias Eschrig explains in simple terms that under additional magnetization orientation originating in the ferromagnet, spin-triplet superconductivity can penetrate a strong ferromagnet with a very small decay. This should enable the survival of the triplet order parameter also in an external magnetic field with very little suppression in low fields as seen here in Figure 5a. It should be noted here that by analyzing and classifying the data of the Hwang group [1] for possible different symmetries of the order parameter of their c-axis oriented $Nd_{0.8}Sr_{0.2}NiO_{2}$ films, Talansev [21] had found that s-wave symmetry is the most likely. He ruled out the p-wave (triplet) symmetry for these c-axis films since they should have a “polar **A**⊥***l***” geometry which gave in the fits to the data an unphysical $2\Delta /kT_{c}$ value of about 1950. In our films however, we made the conjecture that superconductivity of this compound or another Nickelate resides on the surface of the grains which may have a different orientation than c-axis. Therefore, other p-wave geometries are also possible. In particular, the p-wave axial **A**⊥***l*** and polar **A**‖***l*** geometries yield the best fits out of all of Talansev fits to the different symmetries (See Figure 2b,c in his paper [21]). Thus the suggestion of triplet symmetry in our films is quite possible. To further investigate these issues we present in Figure 6 additional current versus voltage curves (IVCs) and conductance spectra of NSNO8an1 on different areas of the film (R8 but also on R2) which are more reminiscent of superconductive tunneling junctions with serial resistance.

Figure 6a shows a full cycle IVCs at 1.8 K and zero field on the R8 area of the NSNO8an1 film under current bias. Increasing current and decreasing current are marked by different colors and the arrows indicate the direction in which the IVC cycle develops. A zoom out to the full current scale is given in the right inset. The prominent features in Figure 6a are the asymmetric voltage jumps, which look symmetric only when the full current cycle is drawn. Comparing the present IVC characteristic to that of a typical Josephson tunnel junction (JTJ) as depicted schematically in the left inset, one finds a striking similarity except for the fact that the Josephson-like current of the central segment of the IVC (the supercurrent $I_{c}$ part) is obviously affected by a serial resistance $R_{\mathrm{serial}}$. When the voltage drop on this resistance is subtracted from the measured voltage, the resulting I vs. V-IR${}_{\mathrm{serial}}$ which is shown in Figure 6b, looks very much like that of a JTJ as depicted again in the inset. The fact that the central $I_{c}$ part is a bit wiggly and not strictly at zero voltage indicates that the $R_{\mathrm{serial}}$ here is not just a simple constant as we assumed to a first approximation. Nevertheless, the similarity of the present IVC to that of a JTJ with serial resistance lend strong support to the notion that the tunneling here originates in superconductivity. Where this superconductivity resides is an open question. We propose that it resides on the surfaces of the Nickelate grains as depicted schematically in Figure 1b for two such grains. If that is the case, transport at low temperatures in our films occurs by tunneling between the superconductive (S) surfaces of the grains via the insulating (I) grain boundaries. This would lead to a network of SIS tunneling junctions connected in series with the additional serial resistance of the normal (N) grains themselves.

To further understand the fine details of the Joshepson-like tunneling here we plot in Figure 6c the conductance spectra of R8 with and without a magnetic field of 4 T. Again, different colors are used for the increasing and decreasing current scans. The V jumps in (a) appear as peaks and dips here. As in Figure 5a where a magnetic field of 1 T was applied at 1.8 K, here a magnetic field of 4 T was applied and in both cases the fields did not change the spectra to within the noise of the measurements. As discussed before, these results are against a spin-singlet superconductive origin of the phenomenon observed, but are in agreement with a possible spin-triplet superconductivity in our films [17]. Figure 6c exhibits in addition to the tunneling behavior between ±0.11 V, a zero bias conductance peak (ZBCP) which could be due to Andreev bound states in a superconductor [22,23]. Using the transport model proposed earlier in Figure 1b for the JTJ structure in our film, we shall try to estimate the energy gap value Δ of the superconducting surfaces of the grains. From Figure 6b one can deduce a value of about 80 meV (half the large voltage jump). Since the film contains several weak-link SIS junctions connected in series between the voltage contacts, we now estimate the number of these junctions. Using the grain size of Figure 1c,d of about 50 nm, and the 2000 nm distance between the voltage contacts, we can estimate the corresponding number of junctions as 40 in this distance. Therefore, one could estimate the value of Δ of a single junction as 2 meV (80/40). This is reasonable, compared to the gaps values of 2–4 meV found in the similar infinite layer compound Nd${}_{1-x}$Sr${}_{x}$NiO${}_{2}$ [8]. Actually, the right inset to Figure 6a reveals additional smaller IVC jumps at ±1.3 V, which are seen as prominent peaks in the conductance spectrum of the inset to Figure 6c. This is similar to the multi quasi-particle branches observed in superconducting Bi-2212 cuprate mesas [24,25,26], so we might be observing two such branches in the present data. It is worth noting in passing, that the resistive $I_{c}$ segment and the first two branches in Figure 2 of Suzuki et al. [26], are very similar to our Figure 6a. In general, the dips in the conductance spectra are due to heating effects when the critical current $I_{c}$ is reached [27,28]. In our case, the intrinsic serial resistance can enhance this heating effect, thus affecting the 2Δ value just as well (an earlier large jump will yield a smaller “gap” value). The conductance spectra of Figure 6c look very different from the previous results of Figure 5b. We therefore show in Figure 6d the more common tunneling behavior as already seen in Figure 5b, but with more resolution and less noise. Here we see that in addition to the continuous narrow tunneling gap structure, we observe the JTJ signature by the sharp peaks and dips. However, this time, the small jumps in the IVC are missing, thus there are only two dips as compared to four dip as in Figure 6c. This means that the JTJ signature is all over our film, but it is not fully developed in R4 of Figure 6d as it is in R8 of (a) and (c) of this figure. The almost linear background signal in Figure 6d is not unusual and is common to other tunneling junctions [29].

### 3.3. Magnetoresistance of the NSNO8 Films

To further shed light on the insensitivity of the conductance spectra to magnetic fields in the present study, we measured the magnetoresistance of the as deposited and annealed NSNO8 films at 2 K. The results are shown in Figure 7. Both films exhibit negative MR of 9% (NSNO8) and 20% (NSNO8an1) at 14 T, except for a slight positive MR between ±2 T in the annealed film. Figure 7d shows comparison of the two films with normalization at zero field. In a recent study by Stupakov et al.. of very similar well oxygenated films of NdNiO${}_{3}$ [30], negative MR similar to what we found in the as deposited film NSNO8, was observed and thoroughly investigated. They showed that the magnetic field Zeeman splitting of the localized states in the films is responsible for the negative MR. Our NSNO8 film is more disordered due to the Sr doping and the lower oxygen pressure used during its deposition process (100 vs. 150 mTorr). Moreover, the films in Ref. [30] were further oxygenated at high temperature in 15 Torr Oxygen on the cool down step from 700 ${}^{\circ }$C after deposition, while our annealing to extract Oxygen was carried out at a much lower temperature of 200 ${}^{\circ }$C. Therefore, it is very likely that oxygen was mainly extracted from the grain boundaries of our films leaving the grains themselves unchanged. Nevertheless, the differences from Ref. [30] did not affect the MR significantly as our 7% MR in NSNO8 at 10 T is similar to their 7–9% MR at 10 T. The oxygen deficient NSNO8an1 is certainly much more disordered, much more resistive and its MR is about twice that (in %) of the NSNO8 film at 14 T as seen in Figure 7d. It therefore follows that while the grains are presumably contributing to the negative MR in quite the same amount in both films, the more insulating grain boundaries in the NSNO8an1 film must contribute the second half of its MR.

A priori, the mechanism of the negative MR in the grain boundaries of our NSNO8an1 film is not necessarily the same as in Ref. [30]. A hint of the existence of a different mechanism is found in the low field positive MR data of Figure 7b,c, which was absent in [30]. The initial small asymmetric hysteresis in this regime is pointing to some ordering in the high magnetic field which then keeps the MR curve symmetric with field cycling between ±14 T without any hysteresis. However, after temperature cycling to about 13 K and back to 2 K, the initial asymmetric hysteresis returned. However, after exposure to high field again, the MR remained symmetric under field cycling as clearly demonstrated in Figure 7c. Apparently, the increased temperature took the film above a magnetic induced ordering transition, and turned it back to its original state. It seems that this ordering of the film is robust (“hard”) at 2 K and if it were a ferromagnetic order, the sample is still below the coercive field even at 14 T. Another phenomenon seen in Figure 7c is the nearly parabolic dependence of the MR between ±2 T. In the inset of Figure 7a this MR data are plotted as conductance 1/R vs. H, and the resulting behavior is parabolic with a dominant term $\propto -H^{2}$. This is typical of flux flow conductance in a superconductor [31,32]. We note that Stupakov et al. [30] did not observe the low field anomaly that we see between ±2 T as in Figure 7c, and we, therefore, cannot compare our results to their results in this regime. The low field MR seen in Figure 7c can not be attributed to weak antilocalization (WAL) either, since WAL functional behavior vs. field near zero field is more like a square root of H [33], rather than nearly parabolic in H as we observe.

Next we discuss whether the present MR results have any bearing on the field independent conductance as seen in Figure 5a and Figure 6c. First, since the magnetic field induced ordering of the film, as seen in the MR data, is robust, it seems likely that unconventional same-spin triplet superconductivity is involved. Spin-singlet superconductivity should have been suppressed by the fields we applied of up to 4 T in the conductance measurements. Second, if superconductivity were present in our films, flux flow would yield negative parabolic-in-field conductance, and we do observe such a behavior between ±2 T as seen in the inset to Figure 7a. Third, the ordering in the MR measurements deduced from the hysteresis in Figure 7c, resets itself after temperature cycling to 13 K and back to 2 K. This could be associated with a superconductive order that vanishes above the transition temperature $T_{c}$ which is less than 13 K. From the above discussion we conclude that superconductivity could be associated with the observed field-induced ordering in the MR results. Moreover, from the huge effect of the low temperature annealing (at 200 ${}^{\circ }$C) on the resistance of the films below 4 K as seen before, this superconductivity must reside in the grain boundary regions at the surfaces of the grains. Therefore, it looks as if the Josephson-like tunneling we observed in Figure 6 is an interface effect of superconductivity with a low volume fraction. Possibly, the infinite layer phase of Nd${}_{0.8}$Sr${}_{0.2}$NiO${}_{2}$ develops on the surface of the grains in our Oxygen deficient films and this is the source of the superconductivity signature that we observed. Local probe measurements such as scanning tunneling spectroscopy (STS) could detect the superconductive energy gap or gaps of this superconductor, being either Nd${}_{0.8}$Sr${}_{0.2}$NiO${}_{2}$ or of a new one. The high film resistance though can make such STS measurements difficult.

## 4. Materials and Methods

The target for the pulsed laser deposition of the films was prepared by mixing stoichiometric amounts of Nd${}_{2}$O${}_{3}$, SrCO${}_{3}$ and NiO, then baking, grinding, and weighing in 5 steps at 930 ${}^{\circ }$C/12 h, 1050 ${}^{\circ }$C/40 h, 1220 ${}^{\circ }$C/24 h, 1250 ${}^{\circ }$C/24 h, and finally pressing and sintering a pellet at 1350 ${}^{\circ }$C/12 h. Most of the weight loss was after the first step where the powder lost most of its CO${}_{2}$, but even in the last step a small weight lost was still found indicating that CO${}_{2}$ was still being removed from the target. For obtaining high quality films of the cuprates it was essential to have Carbon free targets and we believe it is just as important here. For the deposition of the thin films we used third harmonic Nd-YAG laser pulses at 355 nm with 1.5 J/cm${}^{2}$ on the target. The 10×10 mm${}^{2}$ area substrates of (100) SrTiO${}_{3}$ (STO) wafers were clamped to a heater block and deposition was carried out at four different substrate temperatures of 450, 600, 650, and 700 ${}^{\circ }$C (the corresponding heater block temperatures were about 150 ${}^{\circ }$C higher). Deposition was performed under 100 mTorr flow of Oxygen and the deposition rate was about 0.07 nm per pulse. After deposition, the heater was turned off and cooling was under the same Oyxgen pressure flow.

## 5. Conclusions

We have demonstrated that our Oxygen deficient Nd${}_{0.8}$Sr${}_{0.2}$NiO${}_{3-\delta }$ films exhibit Josephson-like tunneling characteristic with serial resistance. Though insulating due to the oxygen poor grain boundaries, the films show signatures of superconductivity in their energy gap opening, in their filling up of this gap with increasing temperature, and in their negative parabolic magneto-conductance typical of flux flow in superconductors. The insensitivity of the conductance spectra to magnetic fields, could be attributed to unconventional same-spin triplet superconductivity. The films can, therefore, be seen as comprised of a network of SIS tunneling junctions residing in the grain boundary regions, on the surface of the grains and the interface regions between them.

## Figures and Tables

**Figure 1 materials-14-07689-f001:**
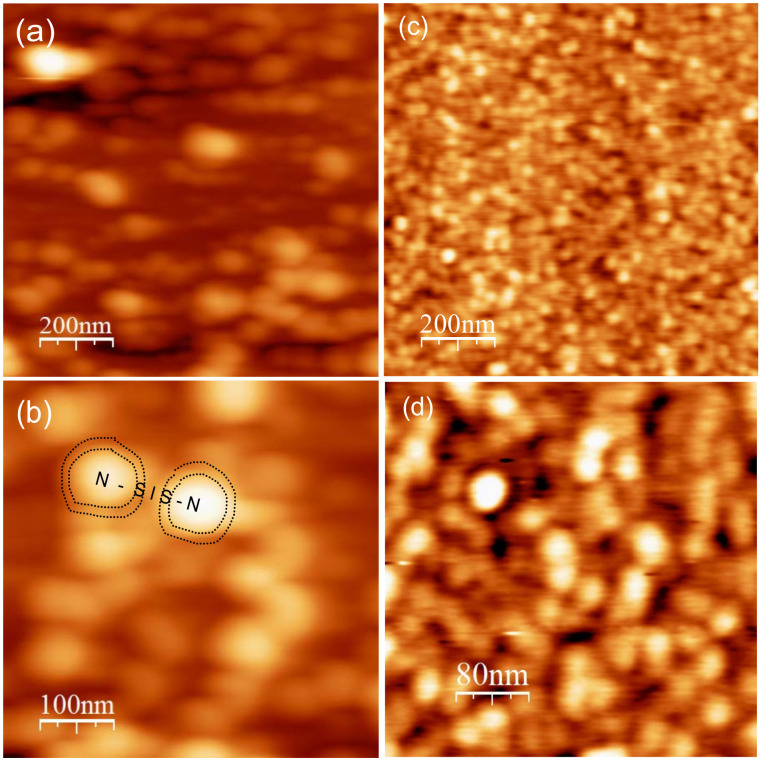
Atomic force microscope (AFM) images of 70 nm thick Nd${}_{0.8}$Sr${}_{0.2}$NiO${}_{3}$ thin films deposited under 650 ${}^{\circ }$C (**a**,**b**) and 600 ${}^{\circ }$C (**c**,**d**) substrate temperatures on (100) SrTiO${}_{3}$ wafers. The smaller grains in (**c**,**d**) are a result of the lower deposition temperature, and the rms roughness accordingly is about 1 nm in (**a**,**b**) and 0.2 nm in (**c**,**d**) which shows that these films are very smooth. A schematic drawing of the conjectured transport model in our films at low temperatures is given in (**b**), where two normal (N) grains (bright) and their superconductive (S) surfaces are marked by the dotted curves encircling these grains. Coupling occurs via the insulating (I) grain boundary, which yields an SIS junction connected in series to the two N grains. It should be noted that the area between the two dotted curves around each grain in this drawing is grossly exaggerated for the purpose of clearly exhibiting this model. The ratio of this area in (**b**) to the encircled grain area is about 1 while in reality it should be much smaller, as a ratio of 1 would lead to percolation of the S areas in the film rendering it wholly superconducting, a phenomenon which we do not observe in the present experiments. As for the structural phase of this S layer, TEM results could be helpful. Unfortunately, in view of the thinness of this surface layer, its irregular and complex geometry, and the huge background of the Perovskite Nd${}_{0.8}$Sr${}_{0.2}$NiO${}_{3}$ grains, the odds of resolving the infinite Nickelate Nd${}_{0.8}$Sr${}_{0.2}$NiO${}_{2}$ structure are very low, and left for future studies.

**Figure 2 materials-14-07689-f002:**
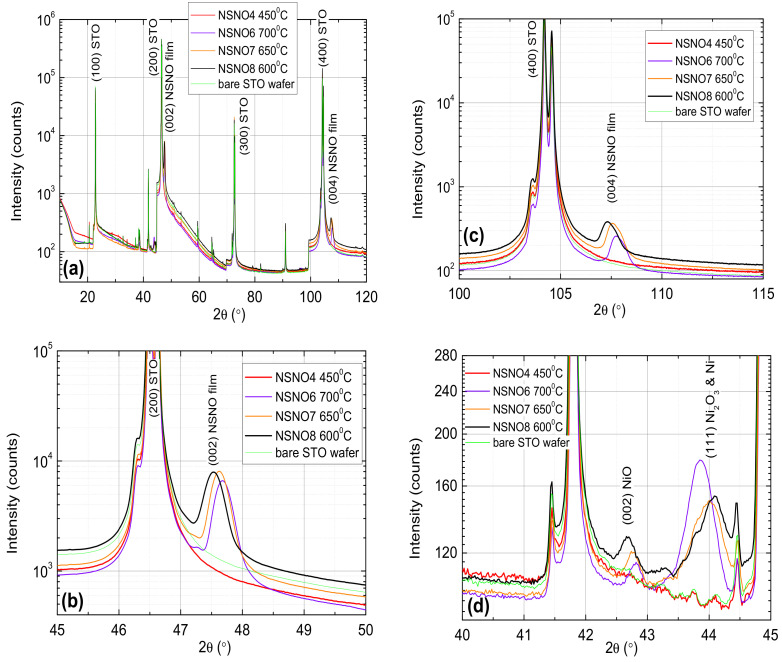
X-ray diffraction θ–2θ scans of 70 nm thick Nd${}_{0.8}$Sr${}_{0.2}$NiO${}_{3}$ thin films deposited under different substrate temperatures on (100) SrTiO${}_{3}$ wafers (**a**). Zoom-in on the (002) and (004) peaks of the films are depicted in (**b**,**c**), respectively. The (001) peak of the films is weak while the (003) peak is absent. Note that both $K\alpha_{1}$ and $K\alpha_{2}$ peaks of the copper X-ray lines at 0.154060 nm and 0.154443 nm are present, thus the STO peaks here appear as doublets with intrinsic angular 2θ split of 0.12${}^{\circ }$ and 0.36${}^{\circ }$ for the (200) and (400) peaks, respectively. The $K\alpha_{2}$ contribution to the data was not removed in order to avoid line shape distortion at the trailing edges of the peaks and for line-widths comparison between the films and the STO wafer. A small amount of minority secondary phases of NiO, Ni${}_{2}$O${}_{3}$, and Ni are also present in our films as can be seen in (**d**). The sharp peaks in (**d**) are due to Tungsten emission lines diffracted of the STO wafer (see the bare STO wafer trace).

**Figure 3 materials-14-07689-f003:**
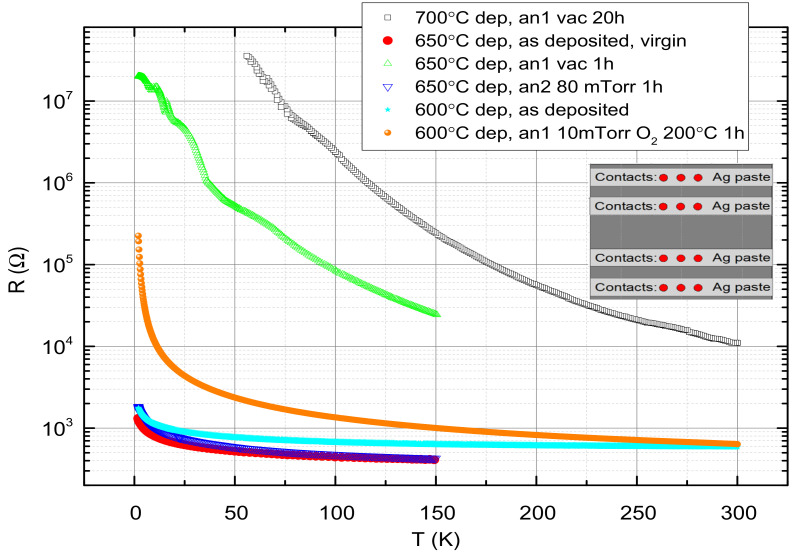
Resistance versus temperature of the three films deposited at 700, 650, and 600 ${}^{\circ }$C either as deposited (virgin if immediately after deposition) or under different annealing conditions, all at 200 ${}^{\circ }$C (first annealing is marked in the figure by an1 and second annealing by an2). The inset shows a top view of the sample with four Ag paste bands for the contacts. The red circles are the locations on the wafer where the contact tips touch the film. Three (out of ten) sets of four contacts locations are shown. The low temperature limit at 55 K, of the most resistive film was set by the V-limit of the PPMS system.

**Figure 4 materials-14-07689-f004:**
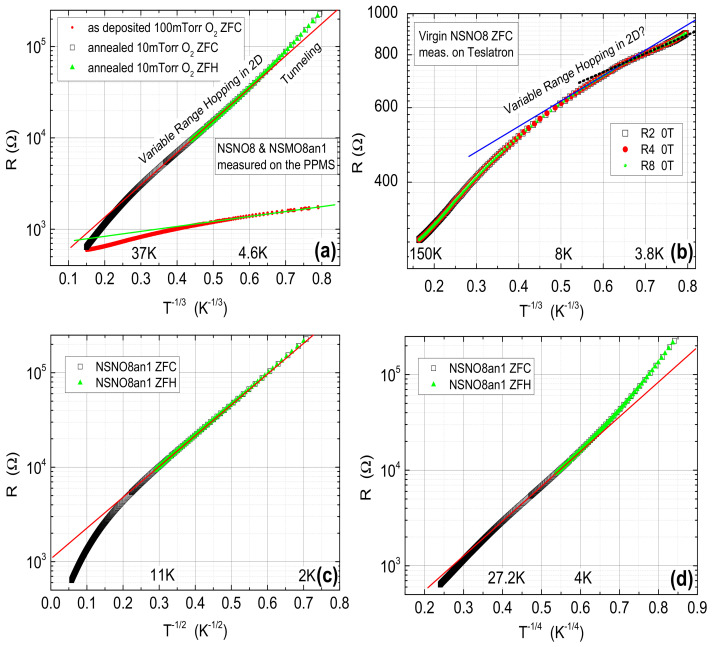
Resistance R on log-scales versus $T^{-1/3}$ (**a**,**b**), $T^{-1/2}$ (**c**) and $T^{-1/4}$ (**d**) where T is the temperature, of the as deposited NSNO8 film under 100 mTorr O${}_{2}$ and 600 ${}^{\circ }$C, and after annealing under 10 mTorr O${}_{2}$ flow at 200 ${}^{\circ }$C for 1 h (NSNO8an1). Measurements were completed on the PPMS (**a**,**c**,**d**)) and on the Teslatron system (**b**). Linear behavior should be observed if transport occurs by variable range hoping (VRH) in 2D (**a**,**b**), in 3D (**d**), or by VRH between conducting grains coupled by the Coulomb interaction (**c**).

**Figure 5 materials-14-07689-f005:**
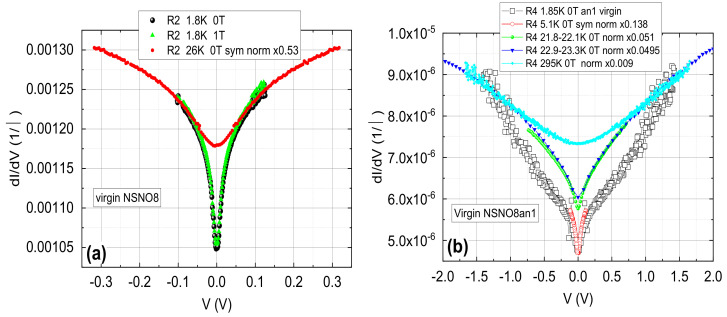
Conductance spectra at various temperatures of the as deposited NSNO8 film (**a**) and the annealed NSNO8an1 film (**b**) on two different areas of the wafer (on the R2 contact in (**a**) and on R4 in (**b**)), with and without a magnetic field of 1 T applied normal to the wafer in (**a**). Different normalization factors are given as “norm ×number” in the figures, and “sym” is short for symmetrized. In (**a**) normalization was performed at 0.1 V and in (**b**) at 1.3 V except for the spectrum at 5.1 K which was visually normalized to the one at 1.85 K.

**Figure 6 materials-14-07689-f006:**
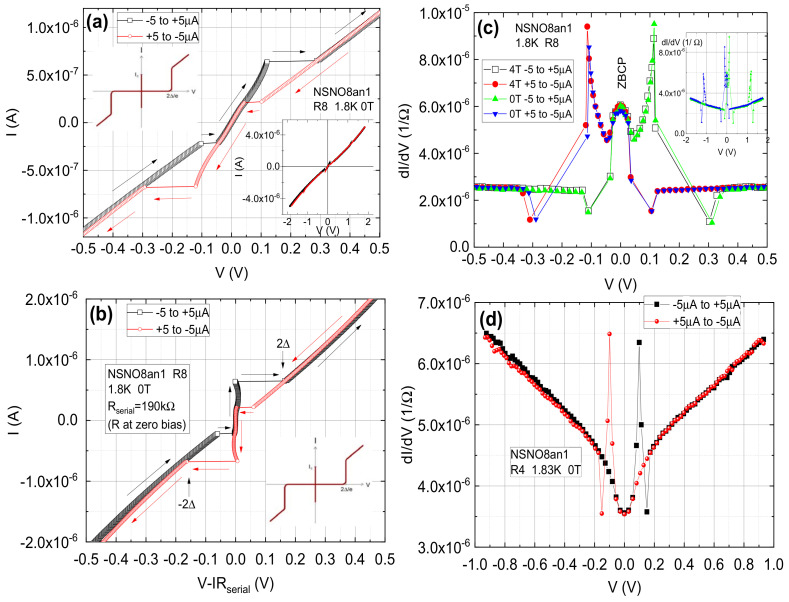
IVC and Conductance spectra at low temperature of the NSNO8an1 film. (**a**) Full cycle IVC of R8 with increasing current bias (black) and decreasing bias (red). The right inset shows a lager bias range with two small additional jumps at about ±1.3 V. Due to the similarity of (**a**) to a Josephson tunneling junction characteristic as in the left inset but with a serial resistance $R_{\mathrm{serial}}$, we plot in (**b**) the same data vs. V-I$R_{\mathrm{serial}}$ which subtract to a first approximation the serial resistance contribution. (**c**) The conductance spectra of R8 with and without magnetic field of 4 T. The variations between the spectra are within the noise of the results. The inset depicts an extended bias spectra. In (**d**) we show for comparison another conductance spectrum on a different area R4 of the film. Clearly, the tunneling behavior as in Figure 5c remains, but there is only a single dip for each current scan direction, unlike in (**c**) where there are two.

**Figure 7 materials-14-07689-f007:**
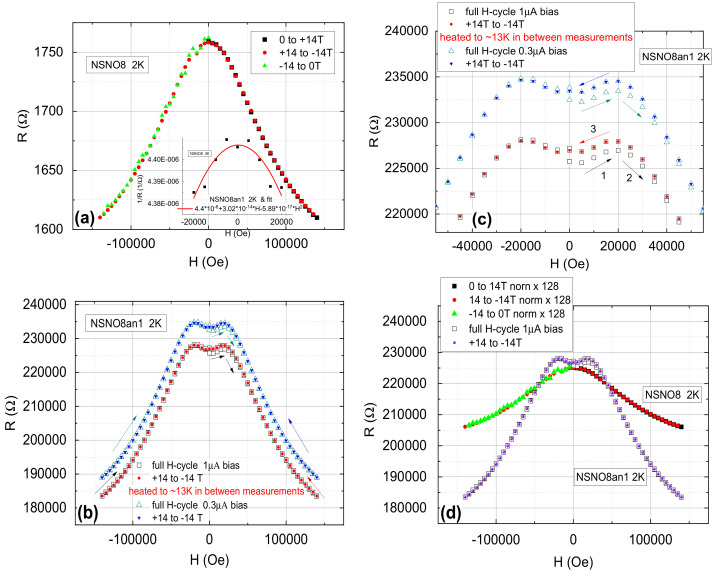
Magnetoresistance of the as deposited NSNO8 film (**a**), and the annealed NSNO8an1 film (**b**,**c**). In (**d**), a comparison between the MR of the two films is given with normalization at zero field. The inset of (**a**) shows conductance at 2 K of NSNO8an1 vs. H in the range ±2 T and a parabolic fit. The NSNO8an1 data of this inset were taken from the 14 to −14 T scan of (**b**).

## Data Availability

The data presented in this study are available on reasonable request from the corresponding author.

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
