# Peer review of "Observation of Josephson-like Tunneling Junction Characteristics and Positive Magnetoresistance in Oxygen Deficient Nickelate Films of Nd0.8Sr0.2NiO3−δ"

_materials, 2021, doi:10.3390/ma14247689_

Round 1

Reviewer 1 Report

Koren et al, have carried electron transport measurements on as-deposited and annealed nickelate film, and claimed the observation of Josephson-like tunneling junction characteristics and positive MR. The topic of infinite nickelate film is hot in the SC community. I am not quite convinced by the present observations before the following points are addressed. Major revision is needed before the publication in Materials.

The following points are still not clear in the manuscript.

  1. The annealing process is vital to yield the oxygen deficient film as the authors claimed. However, no experimental evidences of oxygen deficient were present now.
  2. And what’s the structure of the superconductive surface? They have changed to infinite nickelate film? TEM results may be helpful to resolved what happens after annealing.
  3. I don’t understand why two measuring systems was used in the manuscript.
  4. In line 144, it’s should be “(a) is the most likely model”?
  5. Due to the small superconducting gap in the infinite nickelate film, why the conductance spectra were carried in large voltage range? And about the discussion of superconducting gap in line 204, it’s not clear to claim “a few tens of such junction”. And it’s related the size of grain?
  6. In line 250, the experimental results should be provided in the supplemental information.
  7. Is there thickness dependent? It’s should be sensitive to annealing process. s
  8. Some small comment:

(1) the temperate unite should be “℃”.

(2) In line 61, it should be “reveals”. In line 104, it should be “:” rather than “.”. In line 193, it should be “without”.

(3) In the captain of figure 6, “(c) shows” should be changed to “(c)”.

Author Response

First, thank you for your thorough and valuable report.

Attached is a revised corrections copy of the manuscript with the changes made highlighted in yellow

In the following you will find your comments and criticism and our detailed replies:

Referee 1:

Koren et al, have carried electron transport measurements on as-deposited and annealed nickelate film, and claimed the observation of Josephson-like tunneling junction characteristics and positive MR. The topic of infinite nickelate film is hot in the SC community. I am not quite convinced by the present observations before the following points are addressed. Major revision is needed before the publication in Materials.

The following points are still not clear in the manuscript.

  1. Referee 1: The annealing process is vital to yield the oxygen deficient film as the authors claimed. However, no experimental evidences of oxygen deficient were present now.

Our reply:

The Nickelates structure is very similar to that of the cuprates in the sense that both have Ni oxide and Cu oxide planes with their oxygen easily removed or replenished by proper oxygen poor or rich annealing, respectively. We therefore assumed the same oxygen annealing behavior for both, without providing further experimental evidence for this effect in the Nickelates. Moreover, the orders of magnitude increased resistance of our films following the different oxygen poor annealing processes, indicates oxygen loss, which was reversible on oxygen rich annealing. So there is no question that oxygen gets in an out of our films in the various annealing processes, just as it does in the cuprates.

An explanation of these effects was added in the text (lines 104-108 in the revised manuscript) and a reference of ours from 1989 demonstrating these effects in the cuprates is now added (New Ref. 10).  

2. Referee 1:  And what’s the structure of the superconductive surface? They have changed to infinite nickelate film? TEM results may be helpful to resolved what happens after annealing.

Our reply:

We conjectured that a very thin layer on the surface of the grains is superconducting, whether the infinite Nickelate or another superconductor.  See the schematic model in Fig. 1 (b) where an explanation was added at the end of its caption as to why this layer is very thin. We agree that TEM results could be helpful, but in view of the thinness of this surface layer, its irregular and complex geometry, and the huge background of the Perovskite grains (the NdSrNiO_3 phase), we believe it will be extremely diffucult to resolve the NdSrNiO_2 infinite phase if any, and therefore decided not to pursue TEM measurements in the present study.

Nevertheless, we added some lines of explanation to this effect in the revised manuscript at the end of the caption of Fig. 1. 

3. Referee 1:  I don’t understand why two measuring systems was used in the manuscript.

Our reply:

The answer here is simple. The PPMS system of one system fits all kind of measurements, was not sensitive enough to resolve the fine features of the conductance spectra for which the second Teslatron system was used.

The word “sensitive” was added in the text where the Teslatron system properties are described (line 118 in the revised manuscript).

4. Referee 1:  In line 144, it’s should be “(a) is the most likely model”?

Our reply:

No, since the criterion for choosing (c) in this case is that the linearity of the resistance has the largest resistivity range (10^4 to 2x10^5 Ohms) whereas in (a) it is smaller (3x10^3 to 3.5x10^4 Ohms).

            Nothing was changed in the text since it says exactly this: “If the resistance range                where linearity is observed is the important factor in determining which of the VRH        models applies, then (c) is the most likely model.”

5. Referee 1:  Due to the small superconducting gap in the infinite nickelate film, why the conductance spectra were carried in large voltage range? And about the discussion of superconducting gap in line 204, it’s not clear to claim “a few tens of such junction”. And it’s related the size of grain?

Our reply:

The large voltage range used is due to the fact that we are dealing with a network of junctions connected in series between the voltage contacts. It is highly unlikely to find a single such junction in our contact geometry (using low voltage).

As for the qualitative claim of “a few tens of such junctions”, we now replaced this statement in the text with a more quantitative estimate of the number of junctions between the voltage contacts as follows: “Using the grain size of Fig. 1 (c) and (d) of about 50 nm, and the distance between the voltage contact of about 2000 nm, we obtain an estimated number of 40 junctions in this distance.”  (lines 214-219  in the revised manuscript).

6. Referee 1:  In line 250, the experimental results should be provided in the supplemental information.

Our reply:

We do not understand why this result which is clearly presented in Fig. 7 (c), should be transferred to a supplemental information.

Nevertheless, we added in the text: “…as clearly demonstrated in Fig. 7 (c)”. (line 265 in the revised manuscript)

7. Referee 1:  Is there thickness dependent? It’s should be sensitive to annealing process. S

Our reply:

We tried two more film thicknesses deposited at 630 C (20 and 35 nm) and noticed that the initial resistances at room temperature were too high to carry out the transport measurements. We therefore focused on the 70 nm thick films in the present study, which gave us a manageable dynamic resistance range even after reduction as shown in Fig. 3.  

We added in the text that “thinner films of 20 and 35 nm thickness as compared to the 70 nm thick films used presently, were too resistive to allow for a manageable dynamic resistance range of the transport measurements after reduction as shown in Fig. 3”. (lines 101-104 in the revised manuscript),

8. Referee 1:  Some small comment:

(1) the temperate unite should be “℃”.

(2) In line 61, it should be “reveals”.  In line 104, it should be “:” rather than “.”. In line 193, it should be “without”.

(3) In the captain of figure 6, “(c) shows” should be changed to “(c)”.

Our reply:

All these items were corrected in the revised manuscript.

Reviewer 2 Report

The authors reported a hot topic about nickelate films. And this kind of  Nd0.8Sr0.2 NiO2 are good superconductivity in the infinite layer phase. Furthermore, they discover the oxygen poor situation for this material. The sound of the coupling between superconductive (S) surfaces of the grains in the oxygen poor films via the insulating (I) grain boundaries is very interesting. Also the tunneling junction characteristics are good enough for this work. So I think this paper can be accepted after the language polish.

Author Response

We thank the you for this positive report.

Attached is a revised version of our manuscript as per the other referee reports, with the changes made highlighted in yellow.

Reviewer 3 Report

The work «Observation of Josephson-like tunneling junction characteristics and positive magnetoresistance in Oxygen deficient Nickelate films of Nd0.8Sr0.2NiO3-d» is very well written. Research has been carried out qualitatively. The experimental part is well described. Almost all the necessary literature is cited to understand the point. As a result, the authors demonstrated that our Oxygen deficient Nd0.8Sr0.2NiO3-d films exhibit Josephson-like tunneling characteristic with serial resistance. Though insulating due to the oxygen poor grain boundaries, the films show signatures of superconductivity in their energy gap opening, in their filling up of this gap with increasing temperature, and in their negative parabolic magnetoconductance typical of flux flow in superconductors.

With that said, there are the following comments:

  1. I ask the authors to provide a justification - «The insensitivity of the conductance spectra to magnetic fields could be attributed to unconventional same-spin triplet superconductivity.» (304,305). It is necessary to provide links to works.
  2. Explain how the Neel temperature is related to your system?
  3. Why did the authors take this particular line-up (Sr -20%)?
  4. Your work focuses on oxygen vacancies, but what is the role of Sr in triplet superconductivity that you identified?
  5. Tell me, what is the difference between triplet superconductivity and BCS in your case?

Author Response

First, thank you for your report.

Attached is a correction copy of our revised manuscript with the changes made highlighted in yellow.

In the following you will find your comments and questions with our detailed replies:

Referee 3:

The work «Observation of Josephson-like tunneling junction characteristics and positive magnetoresistance in Oxygen deficient Nickelate films of Nd0.8Sr0.2NiO3-d» is very well written. Research has been carried out qualitatively. The experimental part is well described. Almost all the necessary literature is cited to understand the point. As a result, the authors demonstrated that our Oxygen deficient Nd0.8Sr0.2NiO3-d films exhibit Josephson-like tunneling characteristic with serial resistance. Though insulating due to the oxygen poor grain boundaries, the films show signatures of superconductivity in their energy gap opening, in their filling up of this gap with increasing temperature, and in their negative parabolic magnetoconductance typical of flux flow in superconductors.

With that said, there are the following comments:

  1. Referee 3: I ask the authors to provide a justification - «The insensitivity of the conductance spectra to magnetic fields could be attributed to unconventional same-spin triplet superconductivity.» (304,305). It is necessary to provide links to works.

Our reply:

The possible spin-triplet superconductivity in our films was discussed earlier in the text (lines 162-167), but was not elaborated on as the referee correctly pointed out. We therefore added a new Ref. 18 where an explanation of triplet superconductivity penetration into a ferromagnet with very little suppression is given.

In the text we added “Actually, in Ref. [18] Matthias Eschrig explains in simple terms that under additional magnetization orientation of the ferromagnet spin-triplet superconductivity can penetrate a strong ferromagnet with very small decay. This should enable the survival of the triplet order parameter also in an external magnetic field with very little suppression in low fields as seen here in Fig. 5 (a).” (lines 174-178 in the revised manuscript).

2. Referee 3:  Explain how the Neel temperature is related to your system?

Our reply:

The Neel temperature in bulk NdNiO3 is at about 200 K which coincides with its metal to insulator transition (see Hoda and Yadav, Physica B 491 (2016) 31–36). Since the resistance of our films in Fig. 3 do not show any metal to insulator transition up to 300 K, it is not likely that the Neel temperature affects our films in any significant way.

3. Referee 3: Why did the authors take this particular line-up (Sr -20%)?

Our reply:

Since this is the doping level used in the Nature paper that first found superconductivity in the NdSrNiO system (see Ref. 1). Sr is also the dopant used in the first cuprate high temperature superconductor LaSrCuO_4.

4. Referee 3: Your work focuses on oxygen vacancies, but what is the role of Sr in triplet superconductivity that you identified?

Our reply:

The Sr dopant in the Nickelates plays the same role as in the cuprates, and has no bearing on triplet superconductivity. The cuprates are d-wave superconductors with singlet pairs, unless in combination with a ferromagnet where triplet superconductivity emerges (see Refs. 20 in the revised manuscript). The small amount of ferromagnetic impurities as seen in Fig. 2 (d) here can indicate the role of the triplet order parameter here. See explanation in the text in lines 169-178 of the revised manuscript.

5. Referee 3: Tell me, what is the difference between triplet superconductivity and BCS in your case?

Our reply:

This is now explained in the revised manuscript in lines 174-178. Basically, s-wave (BCS) superconductivity decays rapidly over a short range in a ferromagnet, while triplet superconductivity decays only slightly in a strong ferromagnet under changing magnetization orientation (See Fig. 3 (b) in new Ref. 18).

Round 2

Reviewer 1 Report

The authors have addressed all the points I concerned, and I am satisfied with the current versions.